# LACONIC IMAGE CLASSIFICATION:
# HUMAN VS. MACHINE PERFORMANCE

## ABSTRACT

We propose *laconic classification* as a novel way to understand and compare the performance of diverse image classifiers. The goal in this setting is to minimise the amount of information (aka. *entropy*) required in individual test images to maintain correct classification. Given a classifier and a test image, we compute an approximate minimal-entropy positive image for which the classifier provides a correct classification, becoming incorrect upon any further reduction. The notion of entropy offers a unifying metric that allows to combine and compare the effects of various types of reductions (e.g., crop, colour reduction, resolution reduction) on classification performance, in turn generalising similar methods explored in previous works. Proposing two complementary frameworks for computing the minimal-entropy positive images of both human and machine classifiers, in experiments over the ILSVRC test-set, we find that machine classifiers are more sensitive entropy-wise to reduced resolution (versus cropping or reduced colour for machines, as well as reduced resolution for humans), supporting recent results suggesting a texture bias in the ILSVRC-trained models used. We also find, in the evaluated setting, that humans classify the minimal-entropy positive images of machine models with higher precision than machines classify those of humans.

## 1 INTRODUCTION

Deep neural networks now surpass human-level performance on a variety of specific tasks and metrics relating to visual recognition. In a widely-used yardstick for human-level performance, Russakovsky et al. (2015) estimated that an expert human with prior training can achieve a top-5 classification error rate of 5.1% on a dataset of 1,500 ILSVRC images and 1,000 target classes. Shortly after, He et al. (2015) surpassed human-level performance on the same task achieving 4.9% top-5 error with PReLU-net. Later works would further reduce this error rate, including ResNet (3.6%) (He et al., 2016), Trimps-Soushen (3.0%) (Shao et al., 2016), SeNet (2.3%) (Hu et al., 2019), etc., with contemporary state-of-the-art models more than halving estimated human error for this specific task.

Though such results represent landmark advances, by focusing on classification errors alone, they do not reveal the full story of relative machine performance for image classification. Works on adversarial examples (Dalvi et al., 2004; Szegedy et al., 2014; Nguyen et al., 2015), for instance, establish that human and machine perception diverges greatly for specifically constructed images. Other works have presented bespoke experiments comparing human and machine performance beyond classification errors, presenting evidence for a lack of robustness in the presence of noisy (Russakovsky et al., 2015; Dodge & Karam, 2017; 2019) or incomplete information (Ullman et al., 2016; Wick et al., 2016; Linsley et al., 2017; Ho-Phuoc, 2018; Srivastava et al., 2019), a sensitivity to spatial Fawzi & Frossard (2015); Hénaff & Simoncelli (2016); Xiao et al. (2018); Engstrom et al. (2019) or colour Hosseini & Poovendran (2018); Hosseini et al. (2018) transformations, a lack of generalisation (Geirhos et al., 2018), a bias towards texture (Geirhos et al., 2019), etc., in evaluated models. By applying specific transformations on test images prior to classification, these latter works provide insights into the specific differences in the types of information that humans and machines rely on to perform adequately at this task.

These latter recent works suggest the need for an information-theoretic framework that generalises such issues: a framework within which the performance of classifiers – be they human, machine or other – can be compared and understood, allowing to quantify, in a more fine-grained manner, the

type of information in the input on which a given classifier depends. While previous works address individual or multiple types of information reduction on input images in isolation, a more general framework should allow to combine and compare different types of reduction on test inputs.

In this paper, we propose such a framework, based on the principle of computing and analysing *minimal entropy positive inputs*: inputs with minimal information with respect to yielding correct classification results. The notion of entropy intuitively generalises and allows to compare the relative effects of different reductions on inputs – and their combinations – on classifier performance; such reductions may include, for example, cropping, downsampling and quantisation. The goal in this framework thus shifts from precise classification to *laconic classification*: providing a classifier that minimises the entropy of input(s) required for correct classification. Being based on a continuous notion of entropy rather than a discrete notion of correct/incorrect, this latter goal thus presents a novel challenge beyond minimising classification error—a goal for which state-of-the-art approaches are close to achieving perfect results on datasets like ILSVRC. In fact, existing datasets – such as ILSRVC – can be straightforwardly leveraged for evaluating classifiers under this new goal. Models performing more laconic classification (we currently conjecture) should likewise perform more robustly in practical settings involving incomplete or noisy information capture.

Though the framework we propose can be applied to any classifier for any classification task, herein we first instantiate the framework on the aforementioned problem of image classification. We consider three general operations for reducing the entropy of test images: crop, resolution reduction (downsampling) and colour reduction (quantisation). We then propose two methods for finding the minimal entropy positive images under these reductions for two different types of classifier.

Given a pre-trained machine classifier – where classification can be separated from learning – an input test image, and a set of reduction operations, we use a known search algorithm (Powell, 1964) and apply the given reduction functions to the input image to find (under certain assumptions) the lowest entropy image that the model classifies correctly such that applying any further reduction of entropy leads to incorrect classification. We apply the aforementioned framework to find the minimal entropy images from a sample ILSVRC test-set for state-of-the-art deep neural-network (DNN) models (GoogLeNet, SqueezeNet, ResNet50 and SeNetResNet50), with respect to the three aforementioned reduction operations and their combination. Our experiments show that minimal entropy images are considerably smaller than original images for DNNs; for example, we find that with only 2–6% of the information content of the original test-images (on average) the best performing machine model can still produce a correct classification of the reduced image.

We then consider also *human classifiers* in order to compare their ability to perform laconic classification with DNN models; applying our framework for humans was not trivial since learning cannot be separated from classification (we cannot start with the full input test-image and reduce it since the human will remember the image) and an automatic optimization algorithm was not possible. To cope with this we designed an experiment *reversing* the optimization goal: starting from a void image, the human evaluator may add information incrementally until they believe that a confident classification of the displayed image is possible. We apply this framework with more than 500 human subjects through an online interface. Our results show that the minimal-entropy positive images for humans are considerably smaller in the case of colour and resolution reductions (53–62% the size of the corresponding size for the best performing machine model).

Finally we cross-classified the minimal-entropy positive images among DNNs and humans: given classifier $A$ and $B$, we presented $A$'s minimal-entropy positive images to $B$ and vice versa, computing classification precision. We show that the precision of human classifiers on DNN minimal-entropy positive images was considerably better (0.74 precision in the worst case) than the corresponding results for cross-classification among DNN models (0.02 precision in the worst case for the best model). Furthermore, we find that the DNN models on human images give precisions of 0.03–0.43, depending on the reductions used for the images and the model used for classification.

Our results provide insights into how different classifiers – both machine and human – rely differently on different types of information, providing further evidence to support, for example, prior claims of the lack of robustness to incomplete information relative to humans in such models (Dodge & Karam, 2017), a bias towards texture in ILSVRC-trained models (Geirhos et al., 2019), and so forth. As a more general conclusion, we show that humans are capable of performing (much) more

laconic classification in the evaluated setting than state-of-the-art machine classifiers. We conclude with open challenges regarding the laconic classification of images using DNNs.

## 2   RELATED WORKS

A number of previous works have studied the robustness of image classification for DNNs and humans. Ullman et al. (2016) introduced the notion of a "minimal image" as the smallest region of an image that is still recognisable by a human. They show that a small change in these minimal images can have a drastic effect on recognition. Moreover, Ullman et al. (2016) show that DNNs are unable to accurately recognise human minimal images. To compute the minimal images, they started from the complete image and then iteratively cropped it showing the new cropped image to a different human subject every time. The process stopped when the accuracy of recognizing every crop of the current patch dropped from a threshold. Given the difficulty of computing minimal regions recognisable by humans, their experiments were limited to 10 images (Ullman et al., 2016).

Srivastava et al. (2019) extended the previous work by showing that the sharp drop in accuracy for minimal images can also be observed in DNNs. They defined the notion of "fragile recognition image" as a region of an image for which a small change in size produces a considerable change in accuracy of recognition by DNNs. Srivastava et al. (2019) showed that fragile recognition images are abundant and can occur for several different sizes. The work by Ullman et al. (2016) and Srivastava et al. (2019) can be modelled inside our minimal-entropy framework, which considers not only regions of images (i.e., crop), but also reductions in resolution, colour, and combinations thereof.

Another line of related works study the robustness of DNN image classification with respect to distortions to the input image (Hendrycks & Dietterich, 2019; Geirhos et al., 2018; 2019; Dodge & Karam, 2017; 2019; Fawzi & Frossard, 2015; Hénaff & Simoncelli, 2016; Xiao et al., 2018; Engstrom et al., 2019; Hosseini & Poovendran, 2018; Hosseini et al., 2018). Dodge & Karam (2017; 2019) also study how humans react to these distortions showing that the performance of DNNs is much lower than human performance on distorted images. Hendrycks & Dietterich (2019) constructed a benchmark for image classifier robustness based on several different image distortions. Although all these distortions are designed to make the image recognition task more difficult, not all of them reduce the information in the image. For example, two transformations considered by Geirhos et al. (2018) are image rotation and colour inversion that do not alter the volume of information. Other transformations considered by Geirhos et al. (2018) – such as Eidolon and additive noise – may even increase the image file size. In contrast we currently only consider transformations that reduce the information content of the input image file.

## 3   APPROXIMATING MINIMAL-ENTROPY POSITIVE IMAGES

We now discuss the framework we use for evaluating the laconic classification of images. We first discuss the entropy measure and the reductions applied to images in this work. We then discuss the computation of approximate minimal-entropy positive images (MEPIs) for DNNs and for humans.

### 3.1   ENTROPY MEASURE

Our goal is to find the minimal-entropy positive images (MEPIs) for various classifiers based on various forms of entropy reduction. Although there exist numerous proposed entropy measures (Wu et al., 2013; Larkin, 2016), the analysis of entropy for images is not straightforward due to their dimensionality, particularly in the case of multi-channel images where measures of entropy are further complicated by the conditional and joint entropy that must be measured across the channels. We mitigate this issue by adopting the compressed size of an image using a lossless image encoder as an estimation of image entropy (Larkin, 2016). For these purposes we propose to use the Portable Network Graphics (PNG) encoder – proposed as a successor for the Graphics Interchange Format (GIF) – which combines an initial predictive filtering step to improve compressibility, followed by a DEFLATE compression. The size of PNG images has been shown to outperform more complex measures of entropy (Larkin, 2016), while being widely implemented in a variety of libraries.

## 3.2 ENTROPY REDUCTION

Given an $m \times n$ matrix $\mathbf{A}$, we consider the following forms of entropy reduction:

$\mathbf{A}{\downarrow}_{Q(\kappa)}$ is defined as the (nearest-element) $m \times n$ quantised matrix of $\mathbf{A}$ with quantisation factor $\kappa$, such that $a'_{ij} \coloneqq \arg\min_{v \in \{1,\dots,\kappa\}} (v - \frac{\kappa \cdot a_{ij}}{\max(\mathbf{A})})^2$ for all $a'_{ij}$ in $\mathbf{A}{\downarrow}_{Q(\kappa)}$.

$\mathbf{A}{\downarrow}_{D(\sigma)}$ is defined as the $r \times s$ ($r \le m$, $s \le n$) downsampled matrix of $\mathbf{A}$ with scaling factor $\sigma$ such that $0 < \sigma \le 1$, $\lfloor \sigma m \rfloor = r$, $\lfloor \sigma n \rfloor = s$.

$\mathbf{A}{\downarrow}_{S(\alpha,\beta,\gamma,\delta)}$ is defined as the (contiguous) $p \times q$ submatrix ($p \le m$, $q \le n$) computed from the indices $\alpha, \beta, \gamma, \delta$ ($\alpha < k$, $\beta = \alpha + p$, $\gamma < n$, $\delta = \gamma + q$) such that $\mathbf{A}{\downarrow}_{S(\alpha,\beta,\gamma,\delta)} = \mathbf{A}[\alpha, \alpha + 1, ..., \beta - 1; \gamma, \gamma + 1, \dots, \delta - 1]$; in other words, it removes rows $1, \dots, \alpha - 1, \beta, \dots, m$ and columns $1, \dots, \gamma - 1, \delta, \dots, n$ from $\mathbf{A}$. We use $\mathbf{A}{\downarrow}_{S \triangleright (\alpha)} \coloneqq \mathbf{A}{\downarrow}_{S(\alpha,m,1,n)}$, $\mathbf{A}{\downarrow}_{S \triangleleft (\beta)} \coloneqq \mathbf{A}{\downarrow}_{S(1,\beta,1,n)}$, $\mathbf{A}{\downarrow}_{S \triangledown (\gamma)} \coloneqq \mathbf{A}{\downarrow}_{S(1,m,\gamma,n)}$ and $\mathbf{A}{\downarrow}_{S \triangle (\delta)} \coloneqq \mathbf{A}{\downarrow}_{S(1,m,1,\delta)}$ to denote removing rows/columns (only) from the left, right, top and bottom of the image, respectively.

In the case of images, these reductions correspond, respectively, to colour reduction (parameter: $\kappa$), resolution reduction (parameter: $\sigma$), and crop (parameters: $\alpha, \beta, \gamma, \delta$). We also consider their combination (parameters: $\alpha, \beta, \gamma, \delta, \kappa, \sigma$). We thus have six parameters by which to reduce entropy. We adapt these reductions to multi-channel images in the natural way. In the case of crop, we apply the reduction to all channels separately; however, based on initial experiments with DNNs, rather than remove the rows and columns of the image's channels, we rather replace them with a constant neutral value, which allowed further entropy reduction in positive images by avoiding distortions once images are internally rescaled by the network (i.e., images with narrow crops being "stretched out"). In the case of colour reduction, the nearest quantisation values are computed in the multi-channel case based on Euclidean distance; we further normalise the output values to fill the colour space after the quantisation, choosing equidistant points. In the case of downsampling, the reduction is applied to each channel and maintains the same aspect ratio for square images.

In Appendix A we provide examples of the four types of reduction on an image of a dog.

## 3.3 MINIMAL-ENTROPY POSITIVE IMAGES

Given an input image, a set of entropy reductions, and a classifier, we define a *naive minimal-entropy positive image* to be the smallest (PNG) image derived from the input image using the given reductions for which the classifier provides a *correct* classification (i.e., the top predicted class is correct). To compute such images, we apply reductions in discrete steps. More formally, given a matrix $\mathbf{A}$ and entropy reduction $\star \in \{Q, D, S \triangleright, S \triangleleft, S \triangledown, S \triangle\}$, we define an *atomic reduction step* as $\mathbf{A}{\downarrow}_{\star(\epsilon)}$ such that $\epsilon \coloneqq \min\{x \mid \mathbf{A}{\downarrow}_{\star(x)} \ne \mathbf{A}\}$; i.e., $\epsilon$ is the smallest value that modifies $\mathbf{A}$ under $\mathbf{A}{\downarrow}_{\star()}$. For $\alpha, \beta, \gamma, \delta$ and $\kappa$, the step size is one ($\epsilon = 1$). In the case of $\sigma$, we define $\epsilon$ as the smallest value such that $\lfloor \sigma m \rfloor \ne \lfloor (\sigma - \epsilon)m \rfloor$ or $\lfloor \sigma n \rfloor \ne \lfloor (\sigma - \epsilon)n \rfloor$ (i.e., such that the image is rescaled).

This definition may lead to undesirable results: consider, for example, the case of a classifier making predictions based on the individual (nondescript) pixels of the input image. With varying probability – depending on the number of pixels, images, and classes, the form of classification, etc. – a classifier may for some pixel "guess" the correct class, with that pixel potentially becoming a naive minimal-entropy positive image. To avoid such cases, we add a continuity condition: a *minimal-entropy positive image* (*MEPI*) is then defined to be the smallest image derived from the input image using the given reductions for which the classifier provides a correct classification, and for which there is a continuous path of atomic reduction steps from the input image giving correct classifications.

## 3.4 APPROXIMATING MEPIS FOR DNNS

In the case of DNNs, for a given image and applicable reductions, we apply a search for the value(s) of the parameter(s) that correspond to the MEPI of the image. Our general method begins with the input image, incrementally applying atomic reduction steps while the prediction of the classifier remains correct, backtracking in the case of an incorrect prediction. While relatively simple to implement for reductions with a single parameter, in the case of crop or combined reductions, we have multiple parameters with respect to which we must search; for example, in the case of crop, for

an $m \times n$ image, the number of images to check in the worst-case is potentially $\binom{m}{2}\binom{n}{2}$, i.e., more than 274 billion contiguous sub-images for a $1024 \times 1024$ input image (and this considering crop alone without combination with resolution or colour). Approximation is thus sought.

First, our assumption of continuity helps us to prune the search space: if we reach a set of parameter values for which classification is correct but for which any further atomic reduction is incorrect, we know we can rule out all further reductions from that point. We may still, however, encounter an infeasible search space when considering multiple parameters; to improve performance, we choose to apply a greedy search algorithm: given that the function we wish to minimise is not differentiable but has a fixed number of inputs, we apply Powell's method (Powell, 1964) to find a local minimum.

## 3.5 APPROXIMATING MEPIS FOR HUMANS

Humans operate differently to DNNs and thus require specialised methods to approximate their MEPIs. First, humans cannot separate classification from learning; hence we cannot show the full input image and apply reductions as the human will (of course) remember the full input image. Second, humans require (much) more time per classification; hence the search steps must be adjusted.

In particular, given $\mathbf{A}$, we rather compute a bottom-up approximation of MEPIs for humans. In what follows, given a set of entropy reductions $\mathcal{R} \subseteq \{Q, D, S \triangleright, S \triangleleft, S \triangledown, S \triangle\}$, we use $\mathcal{A}_{\mathcal{R}}$ to denote the set of matrices recursively reachable from $\mathbf{A}$ by some sequence of (atomic) reduction steps from $\mathcal{R}$, including $\mathbf{A}$ itself. Given $\star \in \mathcal{R}$, we can then define $\cdot \uparrow_{\star(\cdot)}$ as the inverse of $\cdot \downarrow_{\star(\cdot)}$ under $\mathcal{A}_{\mathcal{R}}$, such that if $\mathbf{A}' \in \mathcal{A}_{\mathcal{R}}$, then $\mathbf{A}' \uparrow_{\star(x)} = \mathbf{A}''$ if and only if $\mathbf{A}'' \in \mathcal{A}_{\mathcal{R}}$ and $\mathbf{A}'' \downarrow_{\star(x)} = \mathbf{A}'$ for any value $x$ well-defined for $\star$. Now, starting from an initial image $\mathbf{A}_0 \in \mathcal{A}_{\mathcal{R}}$, at each step $i \geq 0$ the human can then either (1) if $\mathbf{A}_i \neq \mathbf{A}$, select an atomic step $\star \in \mathcal{R}$ and increment the entropy of the image by a constant factor $\zeta$ such that $\mathbf{A}_{i+1} := \mathbf{A}_i \uparrow_{\star(\zeta)}$, or (2) select a class for the image from a list if they believe they can correctly classify the image. If the human guesses correctly, this is considered to be an approximate MEPI (for the image, parameters and human); if not, the image is discarded and a fresh image is considered. To mitigate fatigue, larger atomic steps are defined such that no more than 20 are required to reach the input image along a given dimension; i.e., a constant step $\gamma$ is chosen such that $\mathbf{A}$ can be reached from $\mathbf{A}_0$ by applying $\cdot \uparrow_{\star(\gamma)}$ 20 times for each $\star \in \mathcal{R}$. In the case of crop, starting from the central pixel ($\mathbf{A}_0 := \left[a_{\lceil \frac{m}{2} \rceil, \lceil \frac{n}{2} \rceil}\right]$), we offer the option to increment in all directions at once, while in the combined case, we offer the option to increment all parameters at once. This method makes it feasible to approximate MEPIs for humans, but it is important to note that with the limitations of coarser steps and always starting at the central pixel, the MEPIs approximated using this method are more likely to overestimate the entropy required for correct classification.

## 4 EXPERIMENTAL SETTING

We base our experiments on images from the ImageNet Large Scale Visual Recognition Challenge 2012 (ILSVRC2012) (Russakovsky et al., 2015). We select four DNN classifiers for evaluation (all trained on the ILSVRC2012 training set): SqueezeNet (Iandola et al., 2016), GoogLeNet (Szegedy et al., 2015), ResNet50 (He et al., 2016) and SeNetResNet50 (Hu et al., 2019); as discussed in the introduction, the latter two classifiers surpass human-level performance in terms of top-5 classification error on a dataset of 1,500 ILSVRC images and 1,000 target classes.

As acknowledged by Russakovsky et al. (2015), the 1,000 detailed classes of ILSVRC (e.g., COUCAL, SEALYHAM TERRIER) require extensive previous training for humans to perform adequately at classification, not only to visually distinguish the objects, but also to be able to retrieve their label from the 1,000 possible labels. We thus opt to greatly simplify the classes, selecting the following twenty: BEAR, BIRD, CAT, DOG, FISH, FLOWER, FOX, FRUIT, FUNGUS, HIPPOPOTAMUS, INSECT, LION, MONKEY, REPTILE, SHARK, SPIDER, TIGER, VEGETABLE, VEHICLE, WOLF. We select these classes as they should be generally recognisable to humans without prior training; furthermore, we select mostly plants and animals to provide a more challenging classification task, with visually similar classes, such as LION/TIGER, FRUIT/VEGETABLE, DOG/WOLF, INSECT/SPIDER, etc., providing non-trivial cases to visually distinguish. The results of DNN models are then mapped hierarchically to these higher-level classes (e.g., COUCAL $\mapsto$ BIRD, SEALYHAM TERRIER $\mapsto$ DOG). We sample 15 images for each of the 20 classes from the ILSVRC2012 test set for experiments.

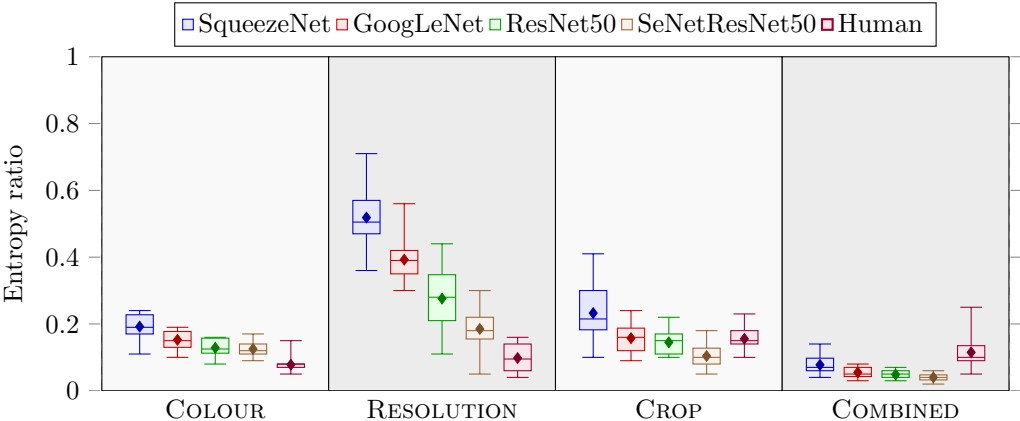

Figure 1: Box-plots of mean entropy ratio for DNN and human MEPIs across classes

## 5 EXPERIMENTAL RESULTS

With our experimental results, we address two main questions: (1) How does the entropy required by DNN and human classifiers compare? (2) How do the classifiers perform in terms of precision for each others' MEPIs? We address these two questions in the following subsections, along with other underlying questions relating to how different image reductions affect individual models, how the goals of laconic and accurate classification correlation, etc.

### 5.1 ENTROPY RATIO IN MEPIS

For the $20 \times 15 = 300$ test images, we first compute the top-down MEPIs for the four DNN classifiers using the method described in Section 3.4. For humans, following the bottom-up method described in Section 3.5, we provide a web interface that – starting with a void image – allows a human user in each step to either increment the image by the given parameters, or select the class for the currently displayed image. In order to achieve many responses, the interface was shared on a university forum as well as on social media. The options and instructions were presented in Spanish, corresponding to the native language of the country in which the university is based. Screenshots of the interface are given in Appendix B. In total, 423 user sessions were logged; of the 1,722 responses obtained (average 4.07 per session), 1,340 (77.8%) resulted in a correct validation and thus a valid MEPI.

In Figure 1 we present the *entropy ratio* for four different settings across five different classifiers; entropy ratio is defined here as the ratio of the (PNG-encoded) size of the original input image versus the size of the extracted MEPI. We take the average entropy ratio for the images of each of the twenty classes. Figure 1 then presents the box-plots – displaying the 1st (min), 25th, 50th (median), 75th, and 100th (max) percentiles with mean marked as a diamond – for the mean ratio across the different classes; to illustrate with an example, in the case of SqueezeNet, considering only the COLOUR experiment, the best class had a mean entropy ratio of 0.11 (bottom whisker), the worst class had a mean entropy ratio of 0.24 (top whisker), etc. We present the DNN models in order of their reported performance for top-5 classification error on the ILSVRC dataset, with SqueezeNet having the highest such error and SeNetResNet50 having the lowest.

From this figure we can draw some high-level observations about the DNN classifiers. First, for the DNN classifiers, the entropy ratio is lowest for the COMBINED experiment (as expected), which offers more avenues by which to reduce the entropy while maintaining a correct classification. Second, the results for DNN classifiers follow the same trend as for performance based on classification error; this suggests that there is a correlation between the goals of laconic classification and precise classification. Third, we see that DNNs are most sensitive to reductions in resolution, which supports the hypothesis that DNNs trained on ILSVRC images are biased towards texture (Geirhos et al., 2019) (with RESOLUTION being the parameter that most affects the ability to distinguish texture). We also see a considerable variation across classes, particularly for RESOLUTION; for space reasons, we provide the detailed results of mean entropy ratio by class in Appendix C.

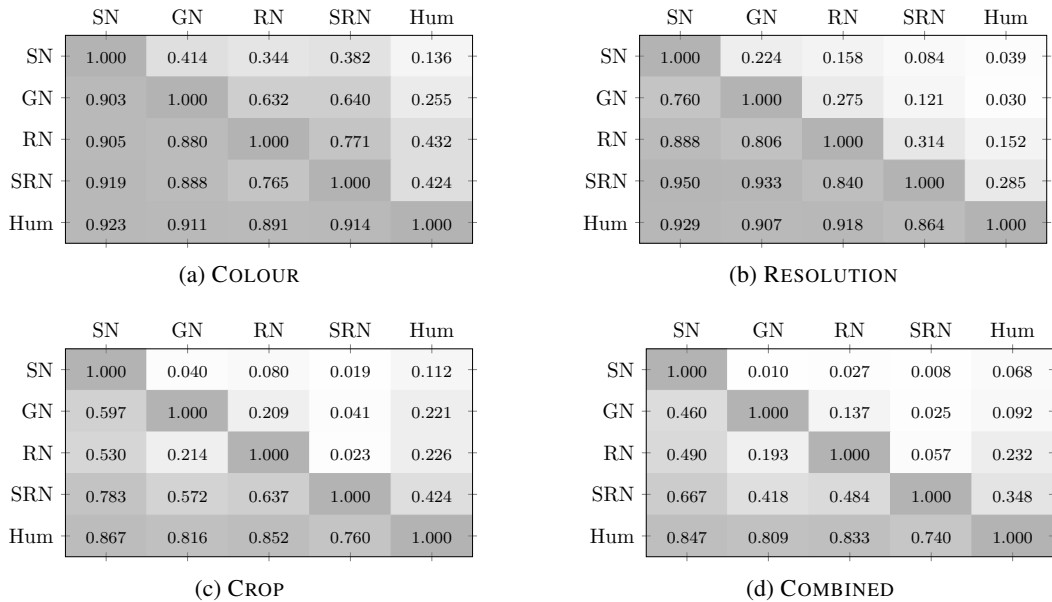

Figure 2: Heatmaps of classification precision on MEPIs for the four types of entropy reduction; rows indicate the classifier, columns indicate the MEPIs classified: SN = SqueezeNet; GN = GoogLeNet; RN = ResNet50; SRN = SeResNet50; Hum = Human

With respect to human classifiers, we note that they are less sensitive to reductions in COLOUR and much less sensitive to reductions in RESOLUTION than all DNNs, and more sensitive to reductions in CROP than some state-of-the-art DNNs. We also see an unusual result, whereby the reduction ratios for COMBINED are not lower than those for (e.g.,) COLOUR; this suggests that the users may have struggled with the interface for the CROP and COMBINED experiments, where multiple options were provided to increment the entropy of the image (versus COLOUR and RESOLUTION, which only permitted increments in one direction). As such, the results for CROP and COMBINED in human classifiers leave an ambiguity: are the relatively poor results of humans due in this case to greater sensitivity to such (multi-parametric) reductions, or because of difficulty using the more complex interface in these cases? We require further experiments to address this ambiguity.

## 5.2 CLASSIFICATION PRECISION FOR MEPIS OF DIFFERENT CLASSIFIERS

We now look at the precision of (top-1) classification across the MEPIs of models, again considering COLOUR, RESOLUTION, CROP and COMBINED. To gather results for human classification, we created a second, simpler, online interface that presents the MEPI of a particular model under a particular reduction and asks the human evaluator to select the class for that MEPI from the list of twenty possible classes. The interface was shared again on a university forum and through social media. We provide a screenshot of this interface in Appendix B. We first ran a control group with 25 trusted users, which logged an aggregate precision of 0.875 with a standard deviation of 0.061; in the open/online evaluation, we then filter user sessions more than two standard deviations from the control mean precision, giving a lower threshold of 0.753. The public evaluation then logged 531 valid user sessions according to the threshold, resulting in 11,588 valid classifications of machine MEPIs (equating to 26.2 classifications on average per session). A comparison between the different groups of humans users is provided in Appendix D.

The results of the precision for cross-classification of MEPIs are summarised in Figure 2. Cells are shaded darker to indicate better performance. Darker rows indicate better classifier performance while darker columns indicate having MEPIs that are easier for other models to classify. First we observe that all classifiers correctly predict (as expected by definition) all of their own MEPIs.

With DNNs ordered by expected performance, we again see the general trend that fewer reported classification errors in ILSVRC again correlate with better precision in the classification of MEPIs,

with, for example, SeNetResNet50 (SRN) having darker rows and lighter columns than other DNNs. We also see good cross-classifier performance for COLOUR and RESOLUTION: given that this are one dimensional reductions, the space of possible images is greatly reduced. On the other hand, DNN models struggle in cross-classification of the MEPIs under CROP and COMBINED: this is perhaps due to the larger search space for these reductions, but also indicates that these MEPIs for DNNs are characteristic of that particular DNN. In general, the MEPIs of SRN are notably more difficult for other models to classify (as could have been expected from Figure 1).

Regarding human classification, we see that in general humans perform best of all classifiers in this experiment, being adaptable enough to classify the MEPIs under all reductions with relatively high precision. The most difficult MEPIs for human are in the CROP and COMBINED configurations for SeResNet50 (SRN), where a relatively high precision of 74–76% correct classifications is still seen. With respect to the previously discussed ambiguity in the results of the previous sub-section, we can thus see that although CROP and COMBINED are more difficult cases for humans, issues with the more complex interface for these cases may explain the relatively poor results of humans in Figure 1 for CROP and COMBINED. Furthermore, we can see that DNNs often struggle to classify the human MEPIs, where the best precision reached was 42% correct classifications for SRN in he case of CROP; Appendix E provides a sample of three human MEPIs not correctly classified by any DNN model, illustrating some of the most difficult such cases for DNN models.

# 6 DISCUSSION

Based on our proposed frameworks for computing minimal-entropy positive images (MEPIs) for both machine and human classifiers, our first results were based on the relative entropy of such images for the different classifiers. From these results we can conclude that state-of-the-art machine models are relatively sensitive (entropy-wise) to reductions in resolution when compared with other forms of reduction based on cropping or colour; such observations appear to independently support previous results indicating a bias in texture for classifiers trained on the ILSVRC dataset (Geirhos et al., 2019). On the other hand, humans are relatively sensitive to the cropping of images, but are much less sensitive than machine classifiers to reductions in resolution; this tends to suggest that humans rely more on form and context rather than textures of small regions of the image. In our second set of results, we performed cross-classification of the MEPIs for different models; we saw that humans greatly outperform machine models for classifying the MEPIs of other models, suggesting a generally more robust aptitude in humans for the laconic classification of images. Of particular interest is that there appears to be a strong correlation between the tasks of laconic and accurate classification (per Figure 1) which suggests that the former may be a new take on the latter: working on laconic classification suggests a range of information-theoretic approaches that could be brought to bear in order to improve classifier accuracy.

Our work has a number of limitations that could be addressed for future work. While the method and interface used for computing human MEPIs in a bottom-up fashion work well for simpler (i.e., one-dimensional) forms of image enhancement, the unexpected result in Figure 1 showing an increase in the size of MEPIs in the combined case suggest that the users had some difficulty using the interface for multi-dimensional settings; given that we filter incorrect images from consideration, we speculate that the users often "overshoot" the MEPI in particular dimensions, finding it difficult to select the particular dimension that is most likely to help them correctly classify the image. Refinements of this idea would be interesting to explore in the future. Furthermore, in order to facilitate non-expert humans to participate, we selected high-level classes, where it would be of interest to try to repeat the results for lower-level classes (though again presenting challenges for the interface).

On the other hand, we have performed experiments for pre-trained, off-the-shelf DNN models. An interesting line of research would be to rather train models specifically for the task of laconic classification. Along these lines, one could consider computing MEPIs in the training set and feeding them (potentially recursively) back into the model; unlike the internal transformations applied in specific DNN models, such a framework for laconic learning could treat a particular model as a black-box. It would further be interesting to explore how models trained in such a manner perform in more traditional classification metrics – error rates, precision, etc. – on the original input images, as well as whether or not they might help to address the observed lack of robustness of state-of-the-art DNNs in the presence of noisy (Russakovsky et al., 2015; Dodge & Karam, 2017; Hosseini et al., 2018;

Dodge & Karam, 2019) or incomplete information (Ullman et al., 2016; Wick et al., 2016; Linsley et al., 2017; Ho-Phuoc, 2018; Srivastava et al., 2019), or their lack of generalisation (Geirhos et al., 2018), or their bias towards texture (Geirhos et al., 2019). Further exploration is needed.

More generally, we view laconic classification as a potentially fruitful avenue to explore: one that is relatively straightforward to implement, can re-use available datasets, connects a number of recent works, provides insights into the performance and robustness of diverse classifiers, and one that may perhaps reveal new challenges and lead to new developments in image classification.

NOTE: *We plan to publish images and other materials online after double-blind review. Please also see appendices for further details.*

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

## A EXAMPLE OF REDUCTIONS AND MEPIS

In Figure 3 we present examples for the four reductions considered; these examples correspond to the MEPIs for an instance of DOG given by SeNetResNet50.

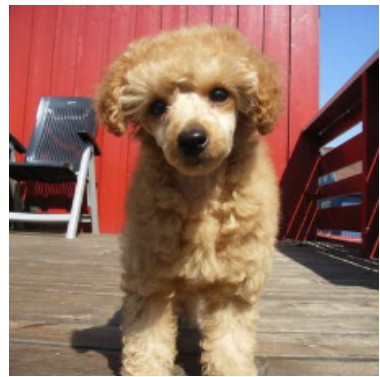

(a) ORIGINAL

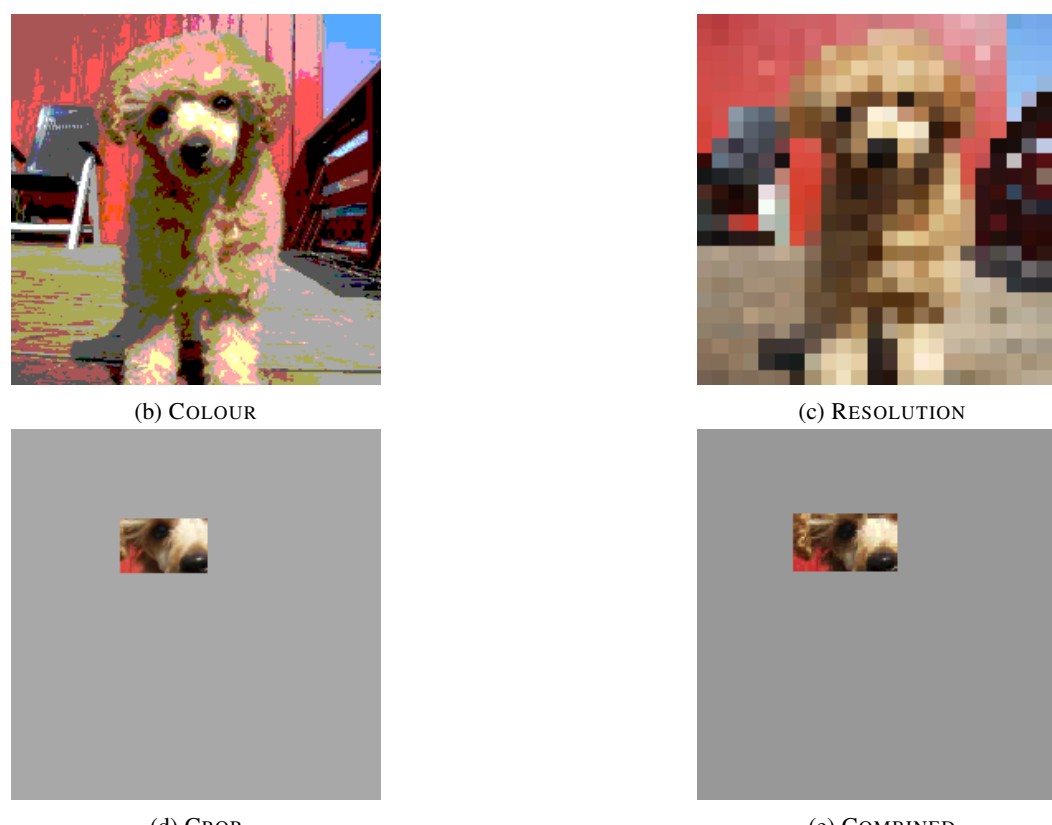

(b) COLOUR

(c) RESOLUTION

(d) CROP

(e) COMBINED

Figure 3: MEPIs for the four types of entropy reduction generated by SeNetResNet50

# B  SCREENSHOTS OF INTERFACES

We present screenshots of the online interfaces used to collect human MEPIs and classifications. As discussed in the paper, the options and instructions were provided in Spanish as the target audience were students of a university whose native language is Spanish. Figures 4–7 show screenshots of the interfaces used to collect human MEPIs for COLOUR, RESOLUTION, CROP and COMBINED. On the other hand, Figure 8 shows a screenshot of the interface to classify machine MEPIs.

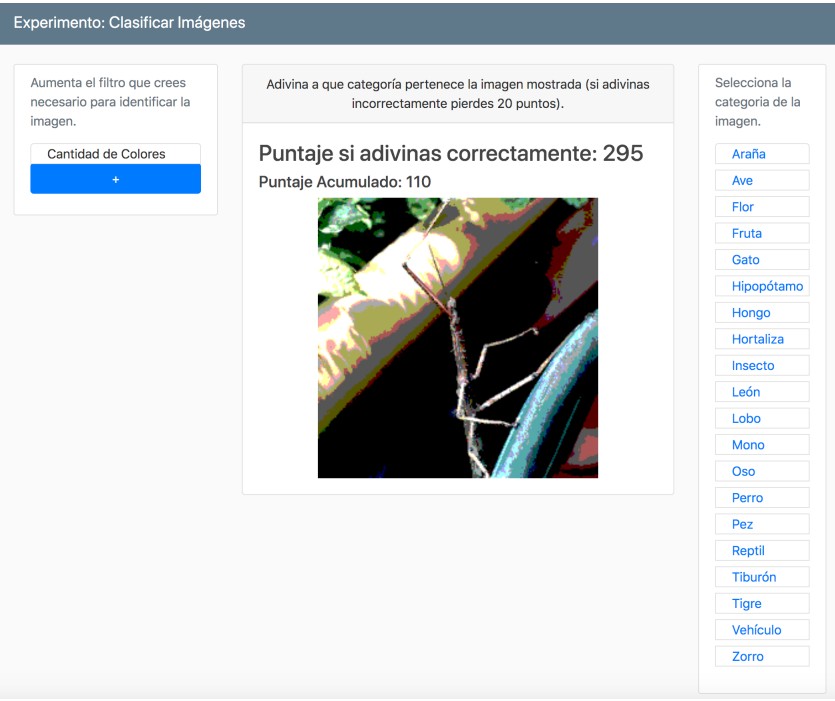

Figure 4: Screenshot of interface for humans to specify COLOUR MEPIs

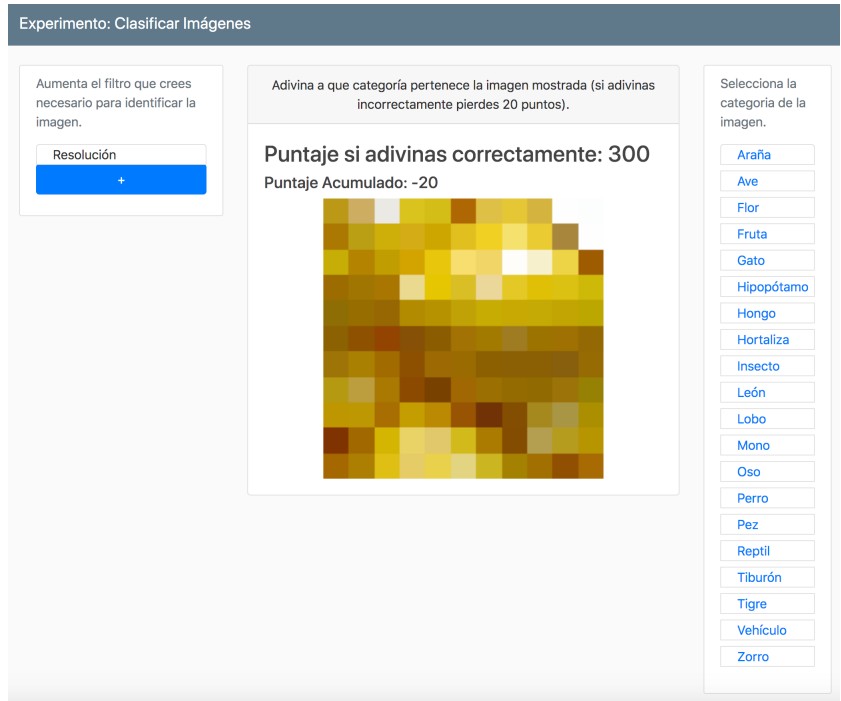

Figure 5: Screenshot of interface for humans to specify RESOLUTION MEPIs

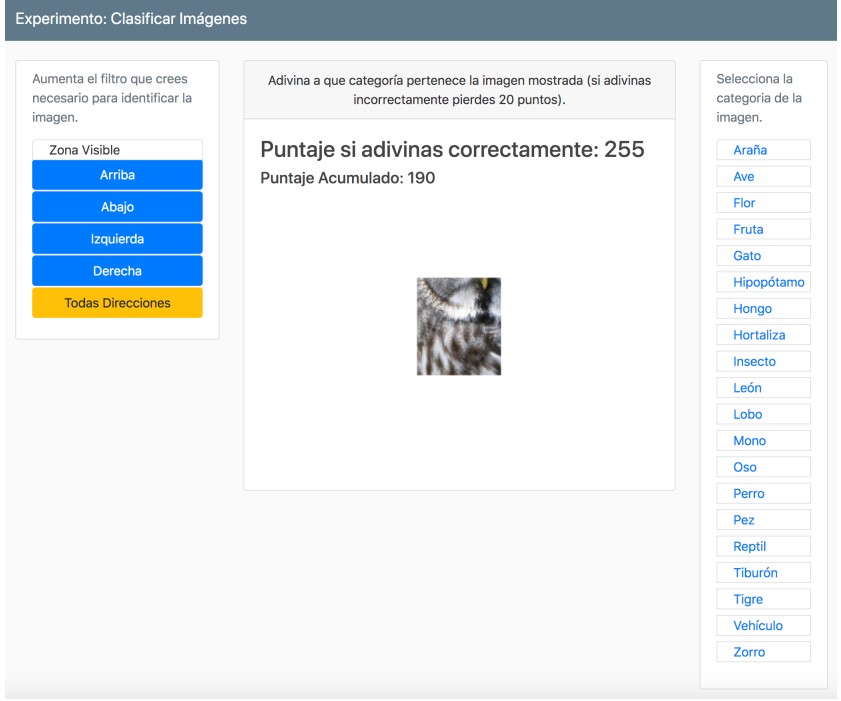

Figure 6: Screenshot of interface for humans to specify CROP MEPIs

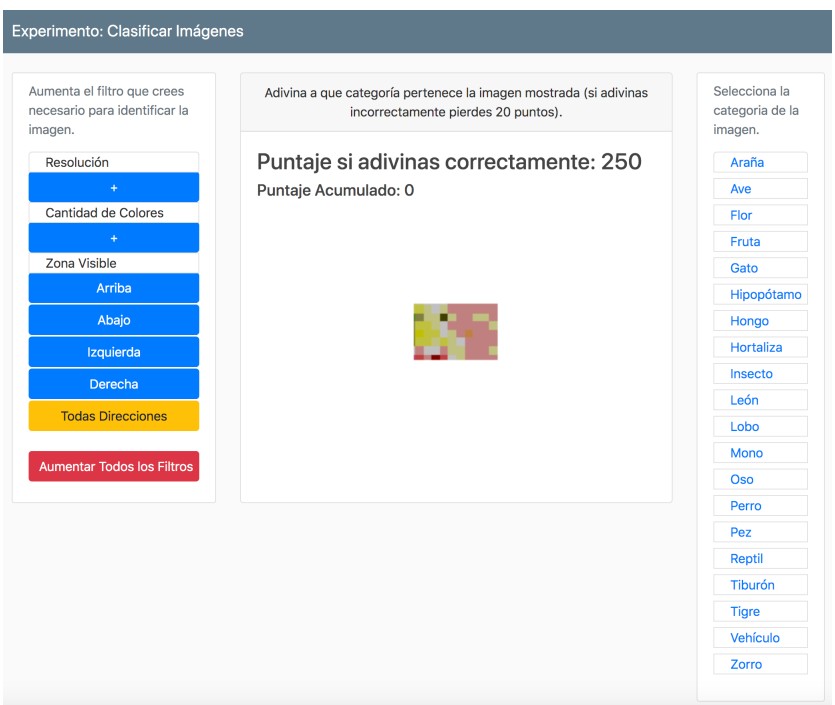

Figure 7: Screenshot of interface for humans to specify COMBINED MEPIs

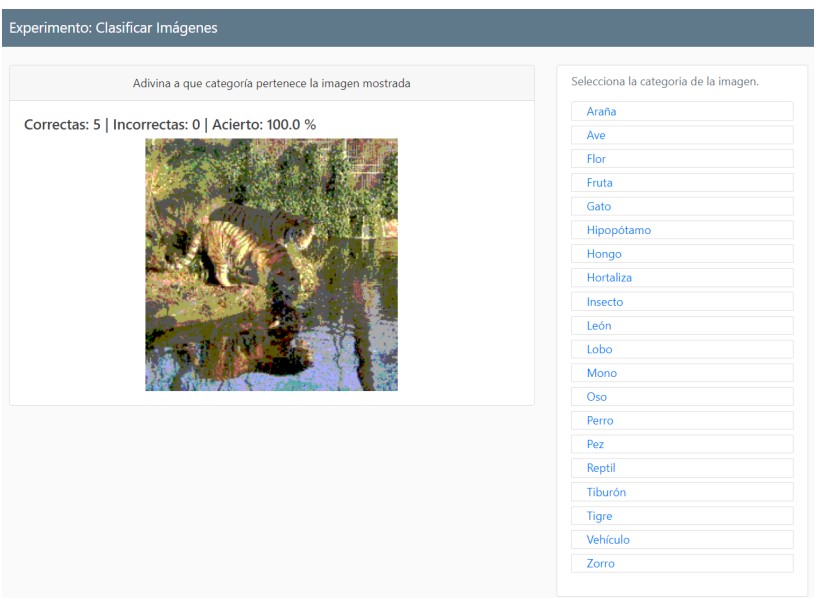

Figure 8: Screenshot of interface for humans to classify machine MEPIs (example using COLOUR)

## C    MEAN ENTROPY RATIO PER CLASS

In Table 1 we present the mean entropy ratio of MEPIs per class.

Table 1: Mean entropy ratio per class for DNN and human MEPIs

| COLOUR | | | | | | | | | |
|---|---|---|---|---|---|---|---|---|---|
| | **bear** | **bird** | **cat** | **dog** | **fish** | **flower** | **fox** | **fruit** | **fungus** | **hippo** |
| SqueezeNet | 0.24 | 0.16 | 0.22 | 0.23 | 0.17 | 0.11 | 0.23 | 0.15 | 0.21 | 0.17 |
| GoogLeNet | 0.18 | 0.14 | 0.17 | 0.16 | 0.15 | 0.1 | 0.18 | 0.13 | 0.19 | 0.15 |
| Resnet50 | 0.16 | 0.12 | 0.12 | 0.12 | 0.12 | 0.08 | 0.16 | 0.11 | 0.16 | 0.13 |
| Se-Resnet50 | 0.17 | 0.11 | 0.12 | 0.14 | 0.13 | 0.09 | 0.14 | 0.15 | 0.14 | 0.11 |
| Humans | 0.1 | 0.07 | 0.07 | 0.06 | 0.08 | 0.09 | 0.08 | 0.07 | 0.15 | 0.08 |
| | **insect** | **lion** | **monkey** | **reptile** | **shark** | **spider** | **tiger** | **vegetable** | **vehicle** | **wolf** |
| SqueezeNet | 0.15 | 0.22 | 0.23 | 0.19 | 0.18 | 0.17 | 0.23 | 0.19 | 0.18 | 0.21 |
| GoogLeNet | 0.12 | 0.19 | 0.18 | 0.15 | 0.13 | 0.12 | 0.17 | 0.15 | 0.12 | 0.17 |
| Resnet50 | 0.09 | 0.16 | 0.15 | 0.1 | 0.13 | 0.09 | 0.16 | 0.14 | 0.15 | 0.12 |
| Se-Resnet50 | 0.11 | 0.12 | 0.15 | 0.12 | 0.12 | 0.1 | 0.13 | 0.14 | 0.09 | 0.12 |
| Humans | 0.07 | 0.07 | 0.08 | 0.08 | 0.05 | 0.06 | 0.08 | 0.09 | 0.08 | 0.05 |
| RESOLUTION | | | | | | | | | |
| | **bear** | **bird** | **cat** | **dog** | **fish** | **flower** | **fox** | **fruit** | **fungus** | **hippo** |
| SqueezeNet | 0.5 | 0.51 | 0.64 | 0.47 | 0.57 | 0.36 | 0.57 | 0.43 | 0.39 | 0.46 |
| GoogLeNet | 0.35 | 0.37 | 0.51 | 0.3 | 0.42 | 0.35 | 0.42 | 0.38 | 0.3 | 0.45 |
| Resnet50 | 0.28 | 0.34 | 0.21 | 0.11 | 0.28 | 0.14 | 0.39 | 0.29 | 0.32 | 0.4 |
| Se-Resnet50 | 0.18 | 0.27 | 0.17 | 0.05 | 0.17 | 0.11 | 0.22 | 0.17 | 0.19 | 0.25 |
| Humans | 0.07 | 0.05 | 0.06 | 0.06 | 0.14 | 0.09 | 0.16 | 0.06 | 0.16 | 0.11 |
| | **insect** | **lion** | **monkey** | **reptile** | **shark** | **spider** | **tiger** | **vegetable** | **vehicle** | **wolf** |
| SqueezeNet | 0.5 | 0.53 | 0.48 | 0.71 | 0.5 | 0.6 | 0.55 | 0.47 | 0.59 | 0.54 |
| GoogLeNet | 0.37 | 0.33 | 0.39 | 0.56 | 0.42 | 0.39 | 0.4 | 0.33 | 0.4 | 0.41 |
| Resnet50 | 0.33 | 0.14 | 0.25 | 0.23 | 0.35 | 0.21 | 0.36 | 0.19 | 0.44 | 0.26 |
| Se-Resnet50 | 0.22 | 0.13 | 0.17 | 0.3 | 0.2 | 0.26 | 0.18 | 0.13 | 0.18 | 0.15 |
| Humans | 0.09 | 0.05 | 0.11 | 0.14 | 0.04 | 0.14 | 0.1 | 0.15 | 0.07 | 0.11 |
| CROP | | | | | | | | | |
| | **bear** | **bird** | **cat** | **dog** | **fish** | **flower** | **fox** | **fruit** | **fungus** | **hippo** |
| SqueezeNet | 0.2 | 0.18 | 0.16 | 0.12 | 0.33 | 0.19 | 0.1 | 0.23 | 0.31 | 0.25 |
| GoogLeNet | 0.12 | 0.12 | 0.12 | 0.09 | 0.22 | 0.16 | 0.1 | 0.12 | 0.22 | 0.16 |
| Resnet50 | 0.18 | 0.12 | 0.11 | 0.1 | 0.16 | 0.17 | 0.11 | 0.11 | 0.2 | 0.17 |
| Se-Resnet50 | 0.09 | 0.08 | 0.08 | 0.06 | 0.14 | 0.13 | 0.08 | 0.05 | 0.12 | 0.13 |
| Humans | 0.18 | 0.16 | 0.13 | 0.19 | 0.23 | 0.14 | 0.16 | 0.14 | 0.21 | 0.18 |
| | **insect** | **lion** | **monkey** | **reptile** | **shark** | **spider** | **tiger** | **vegetable** | **vehicle** | **wolf** |
| SqueezeNet | 0.19 | 0.26 | 0.19 | 0.27 | 0.41 | 0.35 | 0.24 | 0.2 | 0.31 | 0.16 |
| GoogLeNet | 0.18 | 0.18 | 0.14 | 0.24 | 0.19 | 0.17 | 0.16 | 0.16 | 0.2 | 0.1 |
| Resnet50 | 0.17 | 0.11 | 0.15 | 0.13 | 0.22 | 0.15 | 0.15 | 0.11 | 0.17 | 0.11 |
| Se-Resnet50 | 0.11 | 0.11 | 0.1 | 0.13 | 0.18 | 0.12 | 0.09 | 0.08 | 0.1 | 0.1 |
| Humans | 0.18 | 0.14 | 0.18 | 0.16 | 0.14 | 0.1 | 0.14 | 0.14 | 0.1 | 0.11 |
| COMBINED | | | | | | | | | |
| | **bear** | **bird** | **cat** | **dog** | **fish** | **flower** | **fox** | **fruit** | **fungus** | **hippo** |
| SqueezeNet | 0.06 | 0.07 | 0.06 | 0.04 | 0.09 | 0.04 | 0.04 | 0.07 | 0.12 | 0.08 |
| GoogLeNet | 0.05 | 0.04 | 0.05 | 0.03 | 0.08 | 0.05 | 0.03 | 0.05 | 0.08 | 0.06 |
| Resnet50 | 0.05 | 0.04 | 0.05 | 0.04 | 0.06 | 0.05 | 0.04 | 0.03 | 0.06 | 0.06 |
| Se-Resnet50 | 0.04 | 0.04 | 0.04 | 0.02 | 0.06 | 0.04 | 0.04 | 0.02 | 0.05 | 0.05 |
| Humans | 0.12 | 0.07 | 0.09 | 0.05 | 0.15 | 0.17 | 0.09 | 0.1 | 0.25 | 0.11 |
| | **insect** | **lion** | **monkey** | **reptile** | **shark** | **spider** | **tiger** | **vegetable** | **vehicle** | **wolf** |
| SqueezeNet | 0.06 | 0.09 | 0.07 | 0.11 | 0.14 | 0.12 | 0.1 | 0.06 | 0.08 | 0.05 |
| GoogLeNet | 0.04 | 0.06 | 0.05 | 0.07 | 0.08 | 0.07 | 0.07 | 0.05 | 0.05 | 0.04 |
| Resnet50 | 0.04 | 0.04 | 0.06 | 0.05 | 0.07 | 0.05 | 0.05 | 0.03 | 0.06 | 0.04 |
| Se-Resnet50 | 0.04 | 0.04 | 0.04 | 0.06 | 0.06 | 0.04 | 0.02 | 0.03 | 0.03 | 0.04 |
| Humans | 0.11 | 0.1 | 0.1 | 0.2 | 0.08 | 0.09 | 0.12 | 0.14 | 0.07 | 0.09 |

# D  HUMAN PRECISION

In Table 2 we offer a characterisation of the human answers – obtained from online entries and a control group – for the different classes and filters. Used results refer to users in the control group and accepted online entries; Control refers to all users in the control group; Online All refers to all online users; Online Accepted refers to the online users that passed the filter.

Table 2: Mean human precision for the different entries grouping per label and filter

| | bear | bird | cat | dog | fish | flower | fox | fruit | fungus | hippo |
|---|---|---|---|---|---|---|---|---|---|---|
| **COLOUR** | | | | | | | | | | |
| Used | 0.96 | 0.97 | 0.97 | 0.97 | 0.85 | 0.74 | 0.88 | 0.88 | 0.57 | 0.96 |
| Control | 0.98 | 0.99 | 1.0 | 0.99 | 0.87 | 0.75 | 0.89 | 0.89 | 0.58 | 0.98 |
| Online entries | 0.91 | 0.88 | 0.86 | 0.91 | 0.75 | 0.69 | 0.8 | 0.81 | 0.54 | 0.9 |
| Online Accepted | 0.96 | 0.96 | 0.96 | 0.96 | 0.85 | 0.73 | 0.87 | 0.88 | 0.57 | 0.95 |
| | insect | lion | monkey | reptile | shark | spider | tiger | vegetable | vehicle | wolf |
| Used | 0.96 | 0.94 | 0.95 | 0.96 | 0.94 | 0.89 | 0.95 | 0.65 | 0.97 | 0.92 |
| Control | 0.98 | 0.96 | 0.97 | 0.97 | 0.96 | 0.9 | 0.97 | 0.66 | 0.99 | 0.93 |
| Online All | 0.91 | 0.89 | 0.88 | 0.89 | 0.88 | 0.81 | 0.81 | 0.59 | 0.89 | 0.83 |
| Online Accepted | 0.96 | 0.94 | 0.94 | 0.96 | 0.94 | 0.89 | 0.95 | 0.65 | 0.97 | 0.92 |
| **RESOLUTION** | | | | | | | | | | |
| | bear | bird | cat | dog | fish | flower | fox | fruit | fungus | hippo |
| Used | 0.95 | 0.95 | 0.97 | 0.92 | 0.91 | 0.78 | 0.82 | 0.9 | 0.56 | 0.96 |
| Control | 0.96 | 0.96 | 0.99 | 0.94 | 0.92 | 0.79 | 0.84 | 0.92 | 0.57 | 0.98 |
| Online All | 0.9 | 0.89 | 0.89 | 0.84 | 0.86 | 0.7 | 0.77 | 0.83 | 0.51 | 0.85 |
| Online Accepted | 0.95 | 0.94 | 0.97 | 0.92 | 0.91 | 0.78 | 0.81 | 0.9 | 0.56 | 0.96 |
| | insect | lion | monkey | reptile | shark | spider | tiger | vegetable | vehicle | wolf |
| Used | 0.94 | 0.95 | 0.94 | 0.96 | 0.91 | 0.92 | 0.96 | 0.68 | 0.98 | 0.8 |
| Control | 0.96 | 0.96 | 0.96 | 0.97 | 0.93 | 0.98 | 0.69 | 1.0 | 0.82 |
| Online All | 0.85 | 0.87 | 0.85 | 0.87 | 0.83 | 0.83 | 0.9 | 0.63 | 0.88 | 0.72 |
| Online Accepted | 0.94 | 0.95 | 0.94 | 0.95 | 0.91 | 0.92 | 0.96 | 0.67 | 0.97 | 0.8 |
| **CROP** | | | | | | | | | | |
| | bear | bird | cat | dog | fish | flower | fox | fruit | fungus | hippo |
| Used | 0.8 | 0.84 | 0.74 | 0.77 | 0.88 | 0.75 | 0.64 | 0.88 | 0.66 | 0.96 |
| Control | 0.81 | 0.85 | 0.75 | 0.78 | 0.89 | 0.76 | 0.66 | 0.9 | 0.68 | 0.98 |
| Online All | 0.71 | 0.77 | 0.68 | 0.72 | 0.8 | 0.67 | 0.57 | 0.8 | 0.61 | 0.86 |
| Online Accepted | 0.8 | 0.84 | 0.74 | 0.77 | 0.87 | 0.75 | 0.64 | 0.87 | 0.66 | 0.96 |
| | insect | lion | monkey | reptile | shark | spider | tiger | vegetable | vehicle | wolf |
| Used | 0.78 | 0.93 | 0.86 | 0.87 | 0.91 | 0.69 | 0.96 | 0.65 | 0.98 | 0.79 |
| Control | 0.79 | 0.94 | 0.87 | 0.89 | 0.92 | 0.7 | 0.97 | 0.66 | 1.0 | 0.8 |
| Online All | 0.7 | 0.86 | 0.79 | 0.79 | 0.82 | 0.62 | 0.88 | 0.61 | 0.89 | 0.74 |
| Online Accepted | 0.78 | 0.93 | 0.86 | 0.87 | 0.9 | 0.69 | 0.96 | 0.65 | 0.98 | 0.79 |
| **COMBINED** | | | | | | | | | | |
| | bear | bird | cat | dog | fish | flower | fox | fruit | fungus | hippo |
| Used | 0.89 | 0.76 | 0.8 | 0.65 | 0.83 | 0.62 | 0.75 | 0.86 | 0.69 | 0.93 |
| Control | 0.91 | 0.78 | 0.81 | 0.66 | 0.85 | 0.63 | 0.76 | 0.88 | 0.7 | 0.94 |
| Online All | 0.81 | 0.7 | 0.76 | 0.62 | 0.75 | 0.58 | 0.67 | 0.81 | 0.63 | 0.84 |
| Online Accepted | 0.89 | 0.76 | 0.8 | 0.65 | 0.83 | 0.62 | 0.75 | 0.86 | 0.69 | 0.92 |
| | insect | lion | monkey | reptile | shark | spider | tiger | vegetable | vehicle | wolf |
| Used | 0.75 | 0.89 | 0.9 | 0.79 | 0.89 | 0.76 | 0.96 | 0.61 | 0.96 | 0.73 |
| Control | 0.76 | 0.91 | 0.91 | 0.81 | 0.91 | 0.78 | 0.98 | 0.62 | 0.98 | 0.75 |
| Online All | 0.69 | 0.84 | 0.85 | 0.72 | 0.84 | 0.71 | 0.87 | 0.55 | 0.89 | 0.66 |
| Online Accepted | 0.75 | 0.88 | 0.9 | 0.78 | 0.89 | 0.76 | 0.95 | 0.61 | 0.96 | 0.73 |

## E    HUMAN MEPIS NOT CLASSIFIED CORRECTLY BY ANY DNN MODEL

Here we show three examples of human MEPIs not correctly classified by any of the four evaluated DNN models: Figure 9 is of a fox, Figure 10 is of a fungus, while Figure 11 is of a vehicle. All three were generated under COMBINED. The gray border is used to highlight the cropped region.

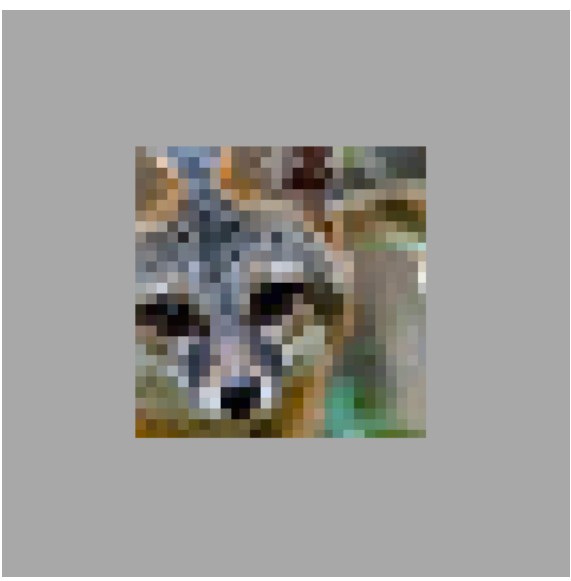

Figure 9: Human MEPI of fox not classified correctly by any DNN model

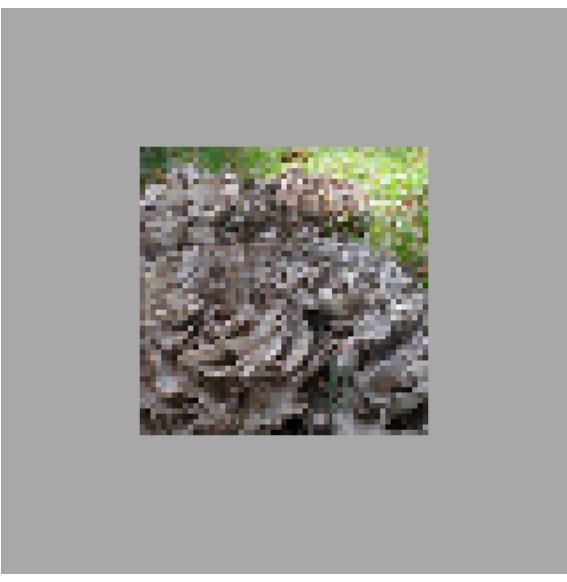

Figure 10: Human MEPI of fungus not classified correctly by any DNN model

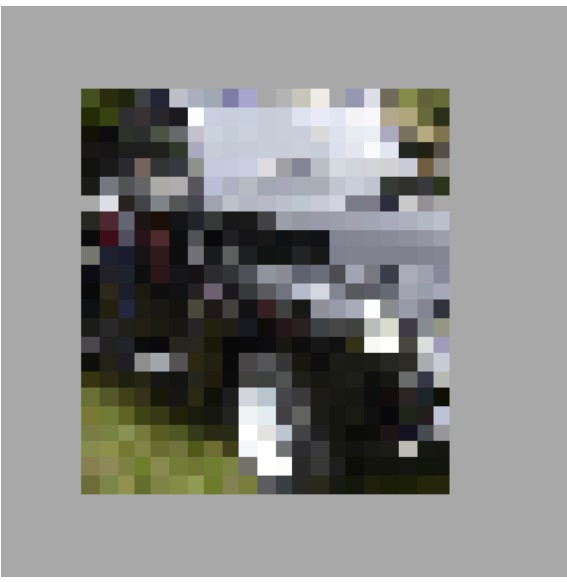

Figure 11: Human MEPI of vehicle not classified correctly by any DNN model

