# OpenReview forum: "Laconic Image Classification: Human vs. Machine Performance"
_ICLR.cc/2020/Conference — Reject_

### Official Review · AnonReviewer1 · 2019-10-14
**Official Blind Review #1**

**Rating:** 1

**Review:**

Summary:
In this empirical study, the authors attempt to identify a minimal entropy version of an image such that the image may be correctly classified by a human or computer. The authors then compare the efficacy of a human and computer to maintain accuracy in the presence of a reduced entropy representation of an image. The authors find that machines are more sensitive to reductions in entropy due to image resolution than humans (as opposed to color or cropping). In addition, the authors find that humans are generally better at identifying minimal entropy images than machines.

1. Corruption results not surprising.

Although the authors offer some intriguing methods, I found the results to not be compelling nor improve our understanding of the relative differences between human and machine perception. While identifying that humans are less sensitive to a reduction in resolution, this result is not terribly surprising given that networks are known to suffer from aliasing artifacts, e.g.

  Geodesics Of Learned Representations
  Olivier J. Henaff & Eero P. Simoncelli
  https://www.cns.nyu.edu/pub/lcv/henaff16b-reprint.pdf

2. Unclear what we learn from the method.

I am not too clear about what specific insights the methods provide in this paper. Estimating the entropy associated with each image corruption -- while interesting -- does not lead to any substantive analysis nor conclusions as far as I can tell. Given the lack of benefit to analyzing the entropy, I am left to really just consider these methods to be image corruptions that downsample the resolution or desaturate the images. These corruptions are not terribly novel with respect to previous work, e.g.

  Generalisation in humans and deep neural networks
  https://arxiv.org/pdf/1808.08750.pdf

  Benchmarking Neural Network Robustness to Common Corruptions and Perturbations
  https://arxiv.org/abs/1903.12261

To summarize, my feedback is the following:

1. Please justify the use of entropy to quantify the distortion. What does entropy provide above and beyond just parameterizing the distortion (e.g. image resolution, color saturation)?

2. Are there other results above-and-beyond sensitivity to image resolution that distinguishes human and machine in these experiments? These results seem to be largely known by just considering corruptions such as low pass filters, etc. presented in the above papers.


Minor Comments:

- The authors should provide a figure with example images in the main text showcasing how each method corrupts an image.

**Experience Assessment:**

I have published in this field for several years.

**Review Assessment: Checking Correctness Of Derivations And Theory:**

I assessed the sensibility of the derivations and theory.

**Review Assessment: Checking Correctness Of Experiments:**

I assessed the sensibility of the experiments.

**Review Assessment: Thoroughness In Paper Reading:**

I read the paper at least twice and used my best judgement in assessing the paper.

---

> ### Author Response · Authors · 2019-11-15
> **Response to Review #1**
>
> ## Geodesics Of Learned Representations
> ## Olivier J. Henaff & Eero P. Simoncelli
>
> We thank the reviewer for the reference, which we have added to the paper along with others looking at invariance to geometric transformations as mentioned by Review #2. (The other references mentioned were already included and indeed are relevant to our work.)
>
> ## 1. Please justify the use of entropy to quantify the distortion. What does entropy provide above and beyond just parameterizing the distortion (e.g. image resolution, color saturation)?
>
> As mentioned, there are various papers that apply reductions, perturbations, distortions, transformations, etc., to images, be it for training or evaluation purposes (as mentioned by the paper and the reviews). These works are of clear importance and were an inspiration for us. However we argue that given all of these works, and the others that will continue to emerge, there is a clear need to generalise these approaches: to be able to incorporate, combine, guide and compare different reductions, perturbations, etc., as part of a common, general, well-founded framework. Our work is then guided by a simple question that we believe underlies all such works, but has not been explicitly identified thus far: what information does a given classifier (machine, human, etc.) require for correct classification? And how can we characterise this information?
>
> As a start, in this paper we propose to adopt an information-theoretic framework based on measuring the entropy of the information required and a method to compute minimal-entropy positive images. We show that this can generalise various image reductions, reproduce existing results from the aforementioned works and can generate novel results for various image reductions. We further use it to compare diverse classifiers.
>
> What we fundamentally wanted to convey with this paper (the "take-home message") is that thinking about the classification problem from an information-theoretic perspective opens up a range of possibilities (some of which are currently being explored from specific angles but without a theory/framework to unite them).
>
> ## 2. Are there other results above-and-beyond sensitivity to image resolution that distinguishes human and machine in these experiments? These results seem to be largely known by just considering corruptions such as low pass filters, etc. presented in the above papers.
>
> Though there are several minor results we could point to, reflecting further on the reviewer’s question, the result we consider of most importance is that there is a clear correlation between accurate classification (low classification error) and laconic classification (requiring less information/entropy in the input) for the classifiers considered. While perhaps not surprising, this result could potentially have interesting consequences. There are thousands upon thousands of works that explicitly optimise for accurate classification, and tens or hundreds of works that explicitly optimise for robustness, but we are not aware of any works that explicitly optimise for laconic classification. Rephrasing the classification task in this way – thinking about how to characterise and reduce the information required by classifiers – is, in our opinion, an interesting new take on an old problem. We understand from this review that the paper fails in some respect to convincingly convey our idea, and perhaps that's something for us to improve upon, but it is, at least for us, an intriguing idea we would be excited to have the opportunity to share and discuss with the community at ICLR.
>
> ## The authors should provide a figure with example images in the main text showcasing how each method corrupts an image.
>
> This is now provided in Appendix A.

---

### Official Review · AnonReviewer3 · 2019-10-21
**Official Blind Review #3**

**Rating:** 6

**Review:**

This paper proposes a method to understand and compare the performance of DNNs classifier, which is different from the precise prediction in the notion of correct/wrong. With approximate minimal-entropy of input images, the classifiers can recognize this image so that different classifiers including human and DNNs will need different reduction methods (cropping, downsampling, color reduction )for the same image to give correct prediction and also will give us different performance in a same test dataset. By comparing the results with human’s and DNNs’, the author claims that it will have more challenges for DNNs in this laconic image classification task than human will have.
For the motivation in this paper, the author tries to propose a new perspective to evaluate the robustness of image classifiers. Especially compared with humans’ understanding, this paper would like to rethink the influence of reduction for this task. However, it is not clear that why three reduction methods the author used can help with understanding the difference between humans and DNNs because when training a DNN for image classification, we usually use these methods to augment our training set, but for humans, it is a totally different story that how to recognize an image.
For the theoretical demonstration, in this paper, the author uses approximating minimal-entropy to quantify the minimal content of an image DNNs or humans need to give correct category. The intuition of this method is suitable. But in section 3, the author didn’t give a clear demonstration of how to compute the entropy reduction in 3.1. I think if it is better to introduce how to measure 3.2 in detail, then 3.1 may be more clear. And it also makes me confused about the atomic reduction step in the last paragraph in 3.1. For the 3.5, I think the authors should focus on how to demonstrate MEPIs for humans more mathematically so that it will be more reliable.
For the experiments, the author tries to answer two questions: 1. How does the entropy required by DNN and human classifiers compare? 2. How do the classifiers perform in terms of precision for each others’ MEPIs? However, the experiments do not provide convincing evidence to existing approaches. First of all, for a single DNN, how different entropy reduction methods influence the classification? Secondly, how different reduction scales in the same model influence the results? At last, the comparison between different models should give a more visualized figure to illuminate the difference. It will be better to provide more ablation study experiments for this paper.


**Experience Assessment:**

I have read many papers in this area.

**Review Assessment: Checking Correctness Of Derivations And Theory:**

I carefully checked the derivations and theory.

**Review Assessment: Checking Correctness Of Experiments:**

I assessed the sensibility of the experiments.

**Review Assessment: Thoroughness In Paper Reading:**

I read the paper at least twice and used my best judgement in assessing the paper.

---

> ### Author Response · Authors · 2019-11-15
> **Response to Review #3**
>
> ## For the motivation in this paper, the author tries to propose a new perspective to evaluate the robustness of image classifiers. Especially compared with humans’ understanding, this paper would like to rethink the influence of reduction for this task. However, it is not clear that why three reduction methods the author used can help with understanding the difference between humans and DNNs because when training a DNN for image classification, we usually use these methods to augment our training set, but for humans, it is a totally different story that how to recognize an image.
>
> DNNs indeed often consider perturbations, transformations, reductions, etc., to augment the training set and increase robustness. Our method is then a general, information-theoretic measure of robustness based on the idea of computing how much information a classifier requires for accurate classifications. As such, it can be used (for example) to evaluate how such augmentations improve classification robustness. Our measure is independent of the form of classifier or training used; although humans and machines use different learning and classification paradigms, they can be compared in such a general framework.
>
> ## For the theoretical demonstration, in this paper, the author uses approximating minimal-entropy to quantify the minimal content of an image DNNs or humans need to give correct category. The intuition of this method is suitable. But in section 3, the author didn’t give a clear demonstration of how to compute the entropy reduction in 3.1. I think if it is better to introduce how to measure 3.2 in detail, then 3.1 may be more clear. And it also makes me confused about the atomic reduction step in the last paragraph in 3.1. For the 3.5, I think the authors should focus on how to demonstrate MEPIs for humans more mathematically so that it will be more reliable.
>
> We agree and have implemented all suggested changes in Section 3. In particular, we moved Section 3.2 before 3.1. We further rewrote the description of the atomic reduction step, and provided mathematical definitions for how MEPIs are computed for humans in Section 3.5 (this turned out not to be trivial and also required additional definitions in earlier sub-sections).
>
> ## For the experiments, the author tries to answer two questions: 1. How does the entropy required by DNN and human classifiers compare? 2. How do the classifiers perform in terms of precision for each others’ MEPIs? However, the experiments do not provide convincing evidence to existing approaches. First of all, for a single DNN, how different entropy reduction methods influence the classification? Secondly, how different reduction scales in the same model influence the results? At last, the comparison between different models should give a more visualized figure to illuminate the difference. It will be better to provide more ablation study experiments for this paper.
>
> We have explicitly added the questions raised by the reviewer as additional questions at the start of the experimental section. Regarding the visual comparison, we were unsure if the reviewer was referring to further plots of results (e.g., for ablation) or rather visual examples of images (MEPIs) for different models. If the reviewer could clarify, we would be happy to look into this for the next version of the paper.

---

### Official Review · AnonReviewer2 · 2019-10-27
**Official Blind Review #2**

**Rating:** 6

**Review:**

The paper proposes and studies a task where the goal is to classify an image that has been intentionally degraded to reduce information content - hence the name "laconic" image classification. The motivation for this task is to compare human and machine performance in a task that deviates from the standard ImageNet setup. To make different content reductions comparable, the authors measure the (approximate) entropy of an image via its PNG-compressed file size. As image transformations, the authors utilize quantization, downsampling, cropping, and a combination of the above. The authors find that convnets with higher accuracy are also more robust to these perturbations, and that humans perform well on the minimum-entropy examples of the networks (but not vice-versa).

Overall I find the comparison of human and machine performance interesting and hence recommend accepting the paper. However, there are multiple directions that could possibly strengthen the core experiment. Hence I only give a weak accept at this point. Concretely, these directions are:

- Do the results in the paper change if a different entropy measure is used (e.g., JPEG compression)?

- As suggested by the authors, training networks to be robust to the "laconic" image perturbations could be an interesting direction. For instance, standard data augmentation with the proposed perturbations would be a relevant baseline.

- To aid replicability and to compare the performance of different human test subjects, it would be interesting to conduct the experiment also on a crowdsourcing platform such as Mechanical Turk (as a complement to, not a replacement for, the university population in the paper).

- Also to add replicability and to make it easier to compare different human accuracy evaluations, it would be good to measure how well the annotators perform on the unperturbed images and on a simple noise transformation (e.g., Gaussian noise).

- It would be good to know how approximate the entropy measures in the paper are, e.g., to understand why humans perform worse in the "combined" perturbation setting.

- How did the results from the control group and the open / online evaluation differ?


In addition, I have the following suggestions for improving the paper:

- For the related work section, the authors may find the following papers on robustness of convnets to distortions interesting:

* Manitest: Are classifiers really invariant?
https://arxiv.org/abs/1507.06535

* Exploring the Landscape of Spatial Robustness
https://arxiv.org/abs/1712.02779

* Spatially Transformed Adversarial Examples
https://arxiv.org/abs/1801.02612

* Semantic Adversarial Examples
https://arxiv.org/abs/1804.00499

- It could be helpful for the reader to see some example images of the different transformations in the main text.

- Section 6: "[...] a bias in texture for images trained on the ILSVRC dataset [...]" - should this be "classifiers" instead of "images"?

- Section 6: "[...] would be interesting to explore in future" - insert "the" before "future"?

**Experience Assessment:**

I have published in this field for several years.

**Review Assessment: Checking Correctness Of Derivations And Theory:**

N/A

**Review Assessment: Checking Correctness Of Experiments:**

I assessed the sensibility of the experiments.

**Review Assessment: Thoroughness In Paper Reading:**

I read the paper at least twice and used my best judgement in assessing the paper.

---

> ### Author Response · Authors · 2019-11-15
> **Response to Review #2**
>
> ## Do the results in the paper change if a different entropy measure is used (e.g., JPEG compression)?
>
> The individual MEPIs (minimal entropy positive images) are not sensitive to the measure of entropy used unless the image reductions (on resolution, colour or crop) increase entropy under the given measure (this does not happen in our setting). The Powell search method is concerned with incrementally reducing the entropy while keeping a correct classification towards a local minimum. A similar process applies for human MEPIs but in reverse. Thus the MEPIs will be the same for any measure that “agrees” that our reductions reduce entropy at each step. The entropy measure becomes important if there are distortions that may or may not increase entropy (which we explicitly do not consider at the moment), or for evaluating laconic classification across different types of reductions/classifiers/images (per Figure 1).
>
> We ruled out lossy compression formats such as JPEG (they are also parameterised). Rather than using lossy compression as an entropy measure, such formats would be an interesting to explore as another form of entropy reduction to explore in our framework (varying the compression ratio).
>
> We also considered “direct” measures of entropy, but did not find any suitable for the scenario; for example, the “delentropy” measure proposed by Larkin (2016), only considers grayscale images, and is outperformed compression-wise by PNG.
>
> Other lossless-compression options to explore might be GIF (largely superseded by PNG), Lossless-JPEG (not widely supported) or WebP-lossless (not widely supported).
>
> ## As suggested by the authors, training networks to be robust to the "laconic" image perturbations could be an interesting direction. …
>
> Indeed this is our priority to continue this work.
>
> ## … it would be interesting to conduct the experiment also on a crowdsourcing platform such as Mechanical Turk …
>
> We agree.
>
> ## … it would be good to measure how well the annotators perform on the unperturbed images and on a simple noise transformation (e.g., Gaussian noise).
>
> We had initially included a Gaussian noise perturbation, but reducing image quality sometimes increased the entropy. We explicitly exclude such perturbations for the moment as they complicate the human MEPI search (they could, however, be considered by the framework, particularly for machine models).
>
> ## It would be good to know how approximate the entropy measures in the paper are, e.g., to understand why humans perform worse in the "combined" perturbation setting.
>
> Without a “standard” established measure of entropy for multi-channel images, we have no baseline against which to measure the level of approximation. On the other hand, the issue of human performance in the combined setting is rather more practical. Starting from a single pixel, the human has several options, such as adding a row of pixels above or below, adding a column of pixels to the left or to the right, increasing the colour, or increasing the resolution. Given that this search space is very large (compared to, e.g., colour where the user advances with one button to improve colours), we also added an option to increase pixels in all directions, as well as to advance in all directions at once, in order to simplify the search in this case. Still the users evidently tend to “overshoot” the MEPI by not knowing which option to select; e.g., increasing resolution slightly might greatly help to recognise the image, but the user, not realising this, may rather choose to continuously expand the crop, increasing the entropy of the MEPI in the “wrong direction”. We acknowledge this limitation in the paper and are unsure how this might be overcome for computing human MEPIs in this specific case (we also experimented with an automated improvement of the image, but ultimately found that this leads to larger MEPIs by taking away too much control from the user, ruling out this option). Still however, the human MEPIs in other cases (particularly colour and resolution) do not have this limitation, nor do the cross-model MEPI classification error results.
>
> ## How did the results from the control group and the open / online evaluation differ?
>
> Most of the online users that were excluded from the results for being too far from the control were essentially users that tried one or two images, gave incorrect answers and gave up on the experiment. We provide mean results for the different user groups in Appendix D.
>
> ## For the related work section ...
>
> We thank the reviewer for these pointers. We agree about their relevance and have referenced them accordingly.
>
> ## It could be helpful … to see some example images of the different transformations in the main text.
>
> We add an example in Appendix A (we could not find space to add them to the main text).
>
>
> We fixed the minor comments (thanks).

---

### Author Response · Authors · 2019-11-15
**Author response to reviews**

We thank the reviewers and the chairs for their consideration and feedback. We have responded to each individual reviewer in turn and have submitted a revised version of the paper.

---

### Decision · Program_Chairs · 2019-12-19

**Decision:**

Reject

**Comment:**

The paper proposes and studies a task where the goal is to classify an image that has been intentionally degraded to reduce information content.
All the reviewers found the comparison of human and machine performance interesting and valuable. However the reviewers expressed concerns and noted the following weaknesses: the presented results are not convincing to support our understanding of the differences between human and machine perception (R1), using entropy to quantify the distortion is not well motivated and has been addressed before (R1), lack of empirical evidence (R2).
AC suggests, in its current state the manuscript is not ready for a publication. We hope the detailed reviews are useful for improving and revising the paper.